# Capturing the Unconscious—The “Psychoanalytic Core Competency Q-Sort”. An Innovative Tool Investigating Psychodynamic Therapeutic Skills

**DOI:** 10.3390/ijerph16234700

**Published:** 2019-11-26

**Authors:** Karoline Parth, Isabelle Wolf, Henriette Löffler-Stastka

**Affiliations:** 1Department of Psychoanalysis and Psychotherapy, Medical University, A-1090 Vienna, Austria; karoline.parth@meduniwien.ac.at; 2University Program for Psychotherapy Research, Postgraduate Unit, Medical University, A-1090 Vienna, Austria; isabelle.wolf@gmail.com

**Keywords:** unconscious, psychoanalytic skills, q-sort, change processes, competence

## Abstract

The Psychoanalytic Core Competency Q-Sort (PCC Q-Sort) is a newly developed empirical research tool that allows for the description and illustration of the ways psychodynamically-oriented psychotherapists work. It provides a simple, straightforward rating procedure utilizing a well-established q-sort method. The present pilot study describes the psychoanalytic core competency items and discusses the development procedure of the instrument as well as statistical analysis of ratings from psychoanalytic sessions, including inter-rater reliability as well as preliminary findings on possible construct validity. Additionally, a factor analysis was performed. Values were assessed by applying the PCC Q-Sort to 30 audio recordings of psychoanalytic sessions. The results of the present study indicate that the PCC Q-Sort is a reliable process research instrument that allows for a detailed investigation of psychotherapy processes in psychodynamic psychotherapies and change processes.

## 1. Introduction

In psychotherapy research, investigations into the specific effects of different psychotherapeutic approaches have become increasingly important. On the one hand, this is due to the fact that there is a growing body of evidence that all major schools, such as cognitive behavioural therapy (CBT), psychoanalysis, and others, are achieving significant improvements with a large number of patients.

A growing body of evidence suggests that therapists significantly contribute to the process and outcome of psychotherapies [1,2,3]. Studies on psychodynamic therapist’s competencies, skills, and interventions summarize several characteristics distinguishing this method from other approaches, such as describing emotional and affective qualities, highlighting the patient’s efforts to avoid certain topics or other attempts to hinder the progress of the treatment, describing and highlighting patterns in the patient’s reports, feelings, actions, and thoughts. Furthermore, stressing past experiences, an emphasis on the current ongoing therapeutic relationship, and as well as focusing on wishes, dreams, or fantasies, have been described as central psychodynamic techniques [4,5,6,7].

Literature on theoretically relevant mechanisms in the process of change in psychodynamic psychotherapy highlights the relevance of the main principles of psychoanalytic theory, namely the relevance of unconscious processes in the development of the psychopathology of the patient, the psychic structure of the patient’s personality, as well as on the relationship dynamics in the therapeutic session [8]. Theoretical and technical concepts on the therapeutic competency necessary to work with these dynamics, such as containment [9,10], interpretation of transference, and countertransference dynamics [11,12,13] as well as working with primitive defenses [14,15], largely inform modern psychoanalytic thinking [16].

However, there is still a lack of research on the specific ways these approaches use their specific competencies to cause change in psychotherapy. In their review, Blagys and Hilsenroth [4] call for studies on the effectiveness of these distinctive variables in regards to patient change in hopes that these will “enable therapists to determine whether these interventions should be adhered to or avoided.” (p. 185)

In order to illustrate the ways in which psychodynamic therapies work (i.e., how they cause symptoms to diminish and personality structure to change) several research groups have listed and systematically described psychoanalytic core competencies on a theoretical basis (e.g., on psychodynamic and psychoanalytic competency: The British Psychological Society’s Centre for Outcomes Research and Effectiveness—CORE) [17]. Others, with a majorly clinical focus, have described in detail how a psychoanalytic session is characterized by specific interventions, stances, and understanding of the therapist (e.g., Working Party on Education of the European Psychoanalytic Federation (WPE) focus group on psychoanalytic competence to practice) [18,19,20].

This enables researchers to better define as well as differentiate psychoanalytic competence from other psychotherapy approaches. Building on these existing efforts as well as on previous psychotherapy process research tools (i.e., the Psychotherapy Q-Sort) [21], the present research aims to bridge the gap between conceptual and clinical investigations to empirical psychotherapy process research. By designing an instrument based on the existing body of research as well as clinical expert opinion, we aim to further possibilities for insight into psychotherapeutic processes. 

This opens up the possibility to highlight the impact of specific psychoanalytic ways of working with patients, namely in focusing on unconscious processes. The PCC Q-Sort aims at contributing to the discussion on the efficacy and working processes of psychodynamic psychotherapies by creating a method to document what goes on in a psychoanalytic session by focusing on the analyst’s actions and stance. With its exclusive focus on psychodynamic methods and interventions, the PCC Q-Sort provides the possibility to describe and investigate the unique qualities of psychodynamic psychotherapies in contrast to many existing measures. For instance, a widely used measure for psychotherapy process evaluation, the Psychotherapy Q-Sort [21], in contrast rather focusses on general differences between psychotherapeutic schools of thought and techniques, and furthermore, describes not only actions by the therapist but also lists patient behaviors. The PCC Q-Sort, on the other hand, takes a different approach in drawing attention to detailed descriptions of the psychoanalytically informed psychotherapist’s actions and rationale. 

In drawing on national and international current concepts on clinical reasoning behind the formulations of core competency, this instrument enables an up-to-date depiction of psychoanalytic techniques and the manner they are employed in clinical practice. In complementing other existing instruments on the therapeutic dyad (i.e., MATRIX) [22], the PCC Q-Sort provides an innovative measure for psychodynamic therapeutic actions and thereby contributes to the scientific empirical investigation of the efficacy of psychotherapy from the perspective of psychotherapeutic competency.

The present study presents results from a pilot study, indicating preliminary positive outcomes on the validity and reliability of the PCC Q-Sort instrument.

### 1.1. The PCC Q-Sort Instrument

The “Psychoanalytic Core Competency” Q-Sort instrument (PCC Q-Sort) is designed to be an empirical research tool that allows the documentation of psychoanalytic work in a psychotherapy session grouped into five areas: 1. items describing the analyst’s overall stance (area 1), 2. items explaining how the analyst refers to and maintains the setting (area 2), 3. items exemplifying how the analyst expresses and makes use of their own personal clinical theory (area 3), 4. items capturing different types of interventions (area 4), and 5. items looking at various kinds of behaviours considered unusual for psychoanalytical approaches (area 5). 

The first area, the psychoanalytic stance, describes the way in which an analyst is able to establish a working atmosphere that allows the patient to associate freely. The second area, the setting, refers to the competencies an analyst displays regarding the maintenance of the analytic setting (e.g., addressing the fact that the patient is late and connecting this to material in the session, timely discussion of breaks, session cancellations). The third area, clinical theory, reflects to which extent psychoanalytic thinking seems to be an influence for the therapist in the session. These items measure whether the therapist seems to draw on psychodynamic concepts on psychopathology, how change is created in the session, and how unconscious processes influence the dynamic in the session and in the patient’s life. This does not specifically refer to individual actions or interventions but rather addresses the overall therapeutic style and attitude of the therapist. The fourth area, interventions, summarizes the different forms of interpretations, clarifications, and interventions that can be expected in a psychoanalytically oriented treatment. The final area, enactments, describes a number of reactions of the analyst that are unusual for their technical and methodological understanding and that express an unfiltered reaction to the patient. The PCC Q-Sort instrument was constructed to be applied to a recording or transcript of a single treatment hour as the smallest unit of observation.

The application of the Q-Sort technique to the analytic hour in its entirety has the advantage of allowing clinical judges to study the material carefully for confirmation of alternative conceptualizations, and to assess the gradually unfolding meaning of events within the analytic hour. It provides a means of objectifying the impressions and formulations derived from a substantial amount of clinical data, while at the same time, summarizing the data through the ordering of a set of statements that describe various aspects of the analytic process [23].

### 1.2. The Q-Sort Method

The Q-Sort technique [23] is a measurement approach with a wide range of possible applications, especially in the area of describing and systemically structuring and describing qualitative data sets. A Q-set is comprised by a set of items, each of which describes a behavioral or interactional property of a situation or an individual. The specific wording of each item can be chosen depending on the situation that is to be studied, which means there is no standard Q-set. Rather, the aim is to provide a measurement technique for varying psychological qualitative contexts and contents [24]. The Q-Sort method follows a positively skewed distribution, where most scores are clustered at the lower end of the curve, with a few very high scores and the majority of the distribution to the lower ends of the curve. 

After listening to the audio recording of an analytic session, the rater then proceeds to order the 27 items, each printed separately on cards to allow for straightforward arrangement. The items are allocated to five groups ranging on a spectrum from “least characteristic” (category 2) to “most characteristic” (category 2). Items that are judged to be either “neutral” or “irrelevant” for the specific session fall in the middle group (category 0). The number of items sorted into each pile (ranging from +2 items in the most characteristic and −2 in the most uncharacteristic categories to 0 in the neutral category) then conforms to the normal contribution mentioned above. This procedure also requires the judges to continuously re-evaluate and sometimes rearrange items along the way in order to avoid negative or positive “halo” effects.

### 1.3. Instrument Construction: Defining Psychoanalytic Core Competencies 

The construction of the PCC Q-Sort was designed to be a multi-phase process. Initially, theoretical concepts on psychoanalytic core competency were reviewed, discussed and grouped. This theoretical basis served as the foundation for the topical areas as well as the major concepts on the manifestation of unconscious processes in the session and specifically the therapist’s actions, stance and behaviors necessary to work with them.

In the second phase, three groups (expert discussion group setting: Psychoanalytic Association Vienna; Department of Psychoanalysis and Psychotherapy, Medical University Vienna; workshop observation setting: Constant Comparative Methods Workshop, Vienna 2015) discussed psychoanalytic core competencies (i.e., what is considered to be essential to psychoanalytic work, what makes psychoanalysis differ from other therapy schools). 

In the final phase, the transcripts from these expert group discussions were then analyzed using qualitative consent analysis, using inductive category formation processes as well as category formation procedures [25]. The items that emerged from the coding and categorizing process of the three expert meetings were then grouped together into thematic clusters and substantiated by theoretical concepts used in contemporary psychoanalytic training institutes.

Using a constant comparative method approach [26], the emerging items were revised in a circular process, in which the expert groups were continuously re-evaluating the clinical relevance of the definitions of the concepts used as well as the rating procedure. Finally, the concepts and approaches discussed by the expert groups were comprised into a list of 27 items.

### 1.4. Applicability

The PCC Q-Sort is constructed and designed to ultimately be able to become a measure for all variations of psychodynamic psychotherapies, providing general enough descriptions and typical as well as untypical items (e.g., area 5 enactments) in order to describe non-standard interventions that might still be part of certain technical approaches. Given the ongoing debate with psychoanalytic experts on the relation between psychoanalytic theory and its implications for technique in regards to modifications of the classical psychoanalytic setting, such as face-to-face twice weekly psychoanalytic psychotherapy, psychodynamic focal therapy, and psychodynamic psychotherapy for specific disorders (i.e., TFP [27]), MBT [28]), it can be expected that in technical considerations, different psychodynamic treatment methods will score with varying results in the various sections of the PCC Q-Sort measure [29,30,31].

However, due to the shared opinion of experts, even though specific interpretative techniques or emphases might differ, the central stance and therapeutic approach of the therapist can be expected to remain stable between these approaches. Therefore, in order to account for these technical differences, the PCC Q-Sort was specifically designed to incorporate techniques and interventions that go beyond the classical or traditional therapist behaviors. This allows for the documentation and classification of different approaches under the umbrella of psychodynamic psychotherapy as well as the comparison of intervention and effect between these subtypes of psychodynamic psychotherapy in regards to the overall treatment process and outcome.

## 2. Methods

### 2.1. Participants

For an initial validation process, audio-recorded psychoanalytic sessions were utilized as data. The sessions were recorded in the course of the Munich Psychotherapy Study [32], a prospective, comparative process-outcome study. All patients were diagnosed with a primary diagnosis of a moderate or severe episode of major depressive disorder (ICD-10 F 32.1/2 or DSM-IV 296.22/23); a recurrent depressive disorder, current episode moderate or severe, without psychotic symptoms (ICD-10 F 33.1/2 or DSM-IV 296.32/33); or a double depression. Exclusion criteria were a diagnosis of bipolar I or II disorder, depression due to physical causes, alcohol or substance dependency, as well as patients who had undergone psychotherapeutic treatment in the course of the last two years or taking any antidepressant medication [32]. All patients included in the present study (N = 10; mean age = 31,2; female = 6, male = 4) received psychoanalytic treatment in an outpatient setting. The treatment was operationalized as a therapy with a usual duration of 240 sessions and a session frequency of three times a week, with the patient lying on the couch. Treatment adherence in the Munich psychotherapy study [32] was documented by recording three sessions at three stages of the treatment (beginning, middle, and ending phase). For the present study, sessions were chosen randomly at an equal rate out of the three recordings in each phase for each patient. 

All in all, the present pilot study included 30 audio-recordings of psychoanalytic sessions from 10 psychoanalytic treatments, ten were from the starting phase of the analysis, ten were from the middle of the treatment process, and ten were taken from the ending phase of the treatment. The study was approved by the ethics committee of the Medical University Vienna (EK Nr. 2169/2013).

### 2.2. Therapists

The study therapists were thoroughly trained and highly experienced. In order to meet the criteria for external validity, treatments in the Munich psychotherapy study [32] were not formally manualized. The practitioner’s mean duration of psychotherapeutic experience was 15 years (range: 6–29 years), the mean age was 47 years (range: 38–56 years). For the purpose of operationalization, psychoanalysis was defined in accordance with the formal definition that it is a “predominantly verbal, interpretative, insight-oriented approach, which aims to modify or re-structure maladaptive relationship representations that lie at the roof psychological disturbance” [33], involving timed interpretations taking into account the therapist–patient relationship as well as possible resistances [34]. 

The 10 psychoanalytic therapists (3 male and 7 female) involved in the study were thoroughly trained according to the standards of the German Psychotherapy Guidelines [35] in their respective training institutes. All therapists were experienced, with many years of psychotherapeutic experience; their mean duration of psychotherapeutic practice was 18 years (range: 8–29 years). The theoretical orientation of the therapists involved was a mixture of Freudian classical and object-relational psychoanalysis, which was assessed using the Therapeutic Attitude Questionnaire (ThAt) [36].

### 2.3. Coders 

All recordings were rated by two female coders, who were graduate and post-graduate students in psychoanalytic psychotherapy training. Coders were trained to the PCC Q-Sort manual and practiced the rating process previous to the actual rating work. Training consisted of studying the rating manual, group discussions on terminology and rating techniques as well as consensus ratings of audio sessions using the present version of the rating manual. In order to reduce rater bias (i.e., preferences to a specific therapy style, other outcomes of process measures), raters were blind to the results of the treatment, and no additional information about the therapists or patients was provided. 

Subsequently to the training process, coders worked independently. Final coding decisions were made after repeated listening to the audio recordings of the individual sessions supported by detailed note-taking. The rating process was partitioned into two steps: in the first step, initial decisions were made regarding positive or negative ratings of individual items, and in the second step, they were then assigned a specific loading (−2, −1, 0, +1, +2). Thereby, adherence to the forced normal distribution of the Q-Sort method was guaranteed.

### 2.4. The PCC Manual and Rating Process

The Manual (more details that appear in the Manual can be obtained upon request from the authors) defines the procedure by which a session audio-recording is rated using a set of 27 items. The PCC Q-Sort Manual describes a set of steps that are followed by the raters; it allows for uncomplicated handling and requires no previous experience in the assessment of psychotherapeutic sessions. The 1st edition of the Manual, used in the present study, is the result of several years of theoretical as well as clinical, expert based research. The items were formed based on a multi-method constant comparative method qualitative research approach. 

### 2.5. The Items

A total of 27 items, which appeared to cover a broad range of the specific properties of psychoanalytic competency with therapists and seemed to be least redundant in describing essential features, were chosen from the material emerging from the content analyses of the expert discussions. Each item was individually reviewed with respect to qualities such as straightforwardness as well as its relevance for psychoanalytic work. Items were rewritten for clarity, and exemplary sample sentences were supplemented. An item was excluded or conflated with a neighboring item if it was considered to contain additional information of value in describing psychoanalytic core competency. Table 1 shows the list of items grouped into five topical areas. 

In the manual, each item is accompanied by detailed rating instructions, for example, “item 13—Interventions are explorative”, has the rating instructions as follows: Place towards characteristic end if A.’s interventions serve to explore the patient’s unconscious internal dynamic and open up space for further meanings and associations (e.g., “I wonder whether what you just told me about your friend stirred up a range of emotions.”). Place towards uncharacteristic end if interventions are rather directive and resemble instructions or explanations (e.g., “As I’ve told you before, your fear of confrontation is connected to your inability to bear anger.”).As the items were largely based on panels of expert discussions, they are intended to depict clinical and theoretical considerations of how psychoanalytic competency is understood within participating in the psychoanalytic institutes and working groups. This allowed for a practically oriented instrument.

### 2.6. Statistical Analysis

Statistical analysis was conducted using IBM SPSS Statistics software (version 24 for Windows, IBM Corporation, Armonk, NY, USA). Inter-rater reliability was determined by using Cohen’s Kappa in order to measure the agreement between both coders and to show to what extent the results are irrespective of the coder [37]. 

Furthermore, a principle component analysis (PCA) with varimax rotation was conducted to explore and visualize the dataset revealing clusters of items, each measuring the same latent characteristics. Varimax rotation, as an orthogonal rotation, was chosen over other rotation options in order to simplify the structure of the factor loading matrix. Varimax maximizes the sum of variances of the squared loadings. Each factor has either large or small loadings of a certain variable [38]. In this process, all 27 items could be included and separated accurately, showing that every item measured exactly what it was supposed to describe with regards to the content. Those items suggesting a “non-analytic technique” were recoded, concerning items 5, 9, 18, 20, 21, 22, 23, 24, 25, 26, and 27. 

## 3. Results

### 3.1. Reliability

The level of significance was set to 5% (*p* < 0.05). Kappa values were generated using crosstabs, calculating the percentage of times both coders actually agreed on the same category (−2, −1, 0, +1, +2) for each item as well as calculating the agreement by chance over all 30 sessions. Using the Cohen’s Kappa coefficient (*κ*), the mean of the overall concordance coefficient was 0.85.

Concordances between each item range from 0.61 to 1.06 showing a substantial to (almost) perfect agreement (<0 = “no agreement”; 0–0.20 = “slight agreement”; 0.21–0.40 = “fair agreement”; 0.41–0.60 = “moderate agreement”; 0.61–0.80 = “substantial agreement”; 0.81–1.00 = “(almost) perfect agreement”) [39]. 

In detail for each item: Item 01: 0.828; Item 02: 0.781; Item 03: 0.824; Item 04: 0.836; Item 05: 0.692; Item 06: 0.785; Item 07: 0.702; Item 08: 0.968; Item 09: 0.763; Item 10: 0.798; Item 11: 1.002; Item 12: 0.884; Item 13: 1.015; Item 14: 0.990; Item 15: 0.910; Item 16: 0.927; Item 17: 0.945; Item 18: 0.817; Item 19: 0.732; Item 20: 0.934; Item 21: 0.976; Item 22: 0.616; Item 23: 0.904; Item 24: 0.916; Item 25: 0.702; Item 26: 1.060; Item 27: 0.735. 

### 3.2. Factor Analysis 

A principle component analysis was conducted as a “preliminary factor analysis”, in which all 27 items could be included. Eight factors could be extracted based on their eigenvalue of at least 1.0. The total variance is 75.720, which means that 76% of variance can be explained by these eight factors. Table 2 lists and describes all actors and their percentage of variance extracted in the principle component analysis.

Factor 1 (explaining 16% of variance) included the following items: listening to the unconscious in the analytic situation, non-normative attitude/neutrality, abstinence, clinical theory reflects influence from other therapy schools, supportive interventions, use of suggestive methods, acting-out. Factor 2 (explaining 11% of variance) included the following items: questions, clarifications, reformulations aiming at making matters conscious, addressing the ‘here and now’, processing/translating material into thinkable material, interventions are ‘out of the frame’, non-interpretive interventions. Factor 3 (explaining 10% of variance) included the following items: receptive listening (evenly suspended attention), self-disclosure, opening up space for thought, timed interventions, use of tools/auxiliaries.

Factor 4 (explaining 8% of variance) included the following items: repairing, psychoanalytic clinical theory is reflected in the technique, interrupting. Factor 5 (explaining 8% of variance) included the following item: connecting different ideas—providing elaborated meaning.

Factor 6 (explaining 8% of variance) included the following items: interventions are explorative, changing the subject. Factor 7 (explaining 8% of variance) included the following items: maintaining the analytic frame, abstract/intellectualized interventions. Factor 8 (explaining 7% of variance) included the following items: translations of theoretical concepts into adequate/patient-specific language, interventions aiming at adding an element to facilitate the unconscious process.

Plotting the eigenvalues in descending order of their magnitude against their factor numbers, the scree plot showed that they level off at factor three and factor six and, therefore, suggest that the optimal number of factors is six. It also showed that the main part of the variance is explained by the first three factors. 

The Kaiser–Meyer–Olkin (KMO) criterion was at 0.234. According to Kaiser this is to be considered an unacceptable value (0–0.49 = “unacceptable”; 0.50–0.59 = “miserable”; 0.60–0.69 = “mediocre”; 0.70–0.79 = “middling”; 0.80–0.89 = “meritorious”; 0.90–1.00 = “marvelous”) [40,41] and, therefore, indicates that the dataset is not suitable for a factor analysis. However, Bartlett’s test of sphericity was significant, suggesting the suitability of the dataset for a factor analysis calculation [42].

## 4. Discussion

The “Psychoanalytic Core Competency” Q-Sort is a novel empirical research instrument based on expert panels and allows for a description of psychodynamic techniques, methods, and clinical behavior in psychoanalytic psychotherapy sessions. This enables a detailed raying of psychodynamic-oriented sessions regarding the therapist’s working patterns. The present pilot study describes the development process and item categories of the PCC Q-Sort and examined its inter-rater reliability and possible hints towards validity. 

All in all, the statistical calculations illustrate that the PCC Q-Sort has a high degree of inter-rater reliability, which showed a substantial to (almost) perfect agreement, suggesting sound applicability of the instrument. In regards to the construct validity, the pilot study could not produce any definite numbers, but the statistical calculations yielded preliminary positive results.

Due to the limited data set of the present study, the Kaiser–Meyer–Olkin criterion was only at 0.234, which speaks against a reliable factor analysis. However, Bartlett’s test of sphericity was significant and indicates the suitability of the dataset for a factor analysis calculation. The principle component analysis, which was provisionally performed, showed that the instrument displays six to eight factors, which means that the items can be grouped into clusters, which have shared latent characteristics that can be clearly delineated.

Keeping that in mind and also considering the high degree of inter-rater reliability, this leads to the conclusion that with a larger data sample, the KMO score will be higher and therefore improving construct validity as well. These preliminary results are, therefore, a first indication for suggesting the instrument’s application for quantitatively investigative psychotherapy process research within the psychodynamic spectrum of treatments. 

The PCC Q-Sort is embedded within a broader context of other process research tools for investigating the role of the therapist in shaping the therapy session and effecting change in the patient–therapist relationship. Based on descriptive findings from several comprehensive working groups, namely the “British Psychological Society’s Centre for Outcomes Research and Effectiveness—CORE” [17] and the “Working Party on Education of the European Psychoanalytic Federation” (WPE) focus group on psychoanalytic competence to practice [18,19,20], the PCC Q-Sort references the competencies listed and displayed by these authors and combines them with theoretical concepts as well as clinical expert opinions and descriptions in order to construct an empirical psychotherapy process measure.

Other existing measures to investigate psychotherapy process have been used as a theoretical and methodological starting point of the PCC Q-Sort (e.g., measuring the psychotherapy style, patient behavior, and session characteristics [24]), and some have previously focused on measuring psychodynamic intervention styles and approaches (e.g., describing the interpretive level [22,43,44]). The PCC Q-Sort complements these existing measures by adding a detailed tool for describing how psychoanalytic therapists use their understanding of unconscious processes, of psychopathology, and of their role in maintaining a specific stance and setting to create a psychoanalytic working process. Thus, it addresses a broad spectrum of interventions, behaviors, and actions by the therapist that are considered to be essential to analytic work. 

This allows researchers for the first time to document and closely investigate how specific psychodynamic treatment approaches contribute to changes within the patients in terms of structural personality changes, symptom reduction, and general improvement of quality of life. Since psychodynamic approaches explicitly consider subtle personality changes to be an essential goal for treatment, it is essential for psychotherapy process research to be able to trace correlations between specific psychodynamic therapist techniques and behaviors and the therapeutic process and outcome of treatments. Furthermore, the application of the PCC Q-Sort to comparative studies between various forms of psychodynamic psychotherapies (i.e., psychoanalysis, psychodynamic psychotherapy) will allow for a more detailed understanding of the specific strengths and advantages of different settings and methods and thereby serve as a helpful addition to theoretical discussions on the merits of various patient groups. 

In the future, larger-scale studies the PCC Q-Sort may allow examinations of fluctuations of specific therapist characteristics—such as the quality of interventions—within different psychodynamic treatment approaches. 

The present pilot study suggests that the PCC Q-Sort is a reliable process research instrument. However, it has limitations: since this is a pilot study introducing the PCC Q-Sort instrument, the present results regarding construct validity have to be considered preliminary results due to the small sample size and presuming better results concerning the construct validity with a larger sample. Furthermore, regarding the development of the PCC items, it has to be taken into account that they are supposed to be based on clinical considerations of experts in the field, which has the advantage that it allows for an instrument that describes how clinicians in these European institutes and working groups understand core competency and how they apply their understanding to their own work. However, this also means that, inevitably, some concepts that might be in use in other institutes might not be included. We hope that future studies would make it possible to extend the existing tool to include an even broader range of clinical concepts and understandings. In addition to the theoretical discussion, we currently plan, on the empirical level, to conduct a confirmatory factor analysis in order to strengthen this first operationalization and see how stable and concise the factors and the items are. 

## 5. Conclusions

Developing new approaches for investigating and assessing psychotherapeutic processes is essential, given the urgent requirement for empirical evidence for the operating principles and outcome of psychodynamic therapy. The PCC Q-Sort joins the existing measures and instruments developed by researchers to explore the manner in which psychoanalytic competency helps the therapist in working with and altering unconscious processes in their patients. 

## Figures and Tables

**Table 1 ijerph-16-04700-t001:** Items of the Psychoanalytic Core Competency (PCC) Q-Sort grouped into five topical areas.

Topical area	Item	Title
Analytic Stance	Nr 1	Attending to unconscious aspects of the analytic situation
	Nr 2	Non normative attitude/neutrality
	Nr 3	Receptive listening (evenly suspended attention)
	Nr 4	Abstinence
Setting	Nr 5	Self-disclosure
	Nr 6	Repairing
	Nr 7	Maintaining the analytic frame (sessions, times, place, relationship configuration)
Clinical Theory	Nr 8	Psychoanalytic clinical theory is reflected in the technique
	Nr 9	Influence from other therapy schools
Interventions	Nr 10	Interventions are intended to bring an idea to consciousness
	Nr 11	Addressing the “here and now”
	Nr 12	Connecting different ideas
	Nr 13	Interventions are explorative
	Nr 14	Opening up space for thought (bear uncertainties, silences)
	Nr 15	Timed interventions
	Nr 16	Translation of theoretical concepts into adequate/patient-specific language
	Nr 17	Processing/translating material into thinkable material (reverie/triangulation)
	Nr 18	Use of tools/ auxiliaries
	Nr 19	Interventions furthering unconscious processes in the session
	Nr 20	Extra-relationship interventions
Enactments	Nr 21	Non-interpretative Interventions
	Nr 22	Abstract/ intellectualized interventions
	Nr 23	Interrupting
	Nr 24	Supportive Interventions
	Nr 25	Changing the subject
	Nr 26	Use of suggestive methods
	Nr 27	Acting-out

**Table 2 ijerph-16-04700-t002:** Factors and their percentage of variance extracted in the principle component analysis.

Factor 1 (explaining 16% of variance) Facilitating a space for unconscious material to come up	▪ Listening to the unconscious in the analytic situation ▪ Non-normative attitude/neutrality ▪ Abstinence ▪ Clinical theory reflects influence from other therapy schools ▪ Supportive interventions ▪ Use of suggestive methods ▪ Acting-out
Factor 2 (explaining 11% of variance) Interventions highlighting the current ucs. process	▪ Questions, clarifications, reformulations aiming at making matters conscious ▪ Addressing the ‘here and now’ ▪ Processing/translating material into thinkable material ▪ Interventions are ‘out of the frame’ ▪ Non-interpretive interventions
Factor 3 (explaining 10% of variance) Maintaining analytic thinking against the pull of the patient’s defense and acting out mechanisms	▪ Receptive listening (evenly suspended attention) ▪ Self-disclosure ▪ Opening up space for thought ▪ Timed interventions ▪ Use of tools/auxiliaries
Factor 4 (explaining 8% of variance) Maintaining a technical stance (third perspective)	▪ Repairing ▪ Psychoanalytic clinical theory is reflected in the technique ▪ Interrupting
Factor 5 (explaining 8% of variance) Connecting different components	▪ Connecting different ideas—providing elaborated meaning
Factor 6 (explaining 8% of variance) Holding onto a train of thought	▪ Interventions are explorative ▪ Changing the subject
Factor 7 (explaining 8% of variance) Speech is reflective of outside circumstances and the patient’s level of thinking	▪ Maintaining the analytic frame ▪ Abstract/intellectualized interventions
Factor 8 (explaining 7% of variance) Facilitating the development of an unconscious thought in the patient	▪ Translations of theoretical concepts into adequate/patient-specific language ▪ Interventions aiming at adding an element to facilitate the unconscious process

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
