# Peer review of "Capturing the Unconscious—The “Psychoanalytic Core Competency Q-Sort”. An Innovative Tool Investigating Psychodynamic Therapeutic Skills"

_ijerph, 2019, doi:10.3390/ijerph16234700_

Round 1

Reviewer 1 Report

The authors make a persuasive argument that it would be useful to be able to perform research on psychodynamic therapy using a natural science tool that groups elements of such theory and practice into measurable and testable categories. I agree and note that there is an important need for demonstrating the effectiveness of psychoanalytic work in outcomes research, which, up until now, has mostly supported manualized CBT treatment modalities. Therefore, I applaud the authors in developing such a measure.

A few additional points:
- The authors need to say more about the experts who developed the original categories.

- The writing needs to be improved. Here are some examples:
- Thereby, in making available an instrument based on the existing 41 body of research that these clinical findings can be used for empirical studies: This is not a full sentence.
- “He” should always be “he or she.”
- Neural” is used instead of “neutral.”
- “Consent analysis” rather than “content analysis.”
- Line 135, Line 163, Line 171, Line 191 run-on sentence.
- Some of the statements are not explained enough, such as Lines 101-103 and all of 104-112, but especially the last sentence about “halo” effect.
- Acronyms should not be used, such as TFP, line 131.
- What is a “constant comparative method qualitative research approach,” line 195. Each such construct needs to be better defined and described.

- It should be indicated, if correct, that “psychodynamic” and “psychoanalytic” are being used interchangeably.

- The paragraph starting on Line 197 does not describe who made those decisions.

The tables are not clear where each factor starts and ends.

Author Response

The authors make a persuasive argument that it would be useful to be able to perform research on psychodynamic therapy using a natural science tool that groups elements of such theory and practice into measurable and testable categories. I agree and note that there is an important need for demonstrating the effectiveness of psychoanalytic work in outcomes research, which, up until now, has mostly supported manualized CBT treatment modalities. Therefore, I applaud the authors in developing such a measure.

A few additional points:
- The authors need to say more about the experts who developed the original categories.

--> Thank you for the suggestion, we added data to describe the experts.

- The writing needs to be improved. Here are some examples: 
- Thereby, in making available an instrument based on the existing 41 body of research that these clinical findings can be used for empirical studies: This is not a full sentence.
- “He” should always be “he or she.”
- Neural” is used instead of “neutral.”
- “Consent analysis” rather than “content analysis.”
- Line 135, Line 163, Line 171, Line 191 run-on sentence.
- Some of the statements are not explained enough, such as Lines 101-103 and all of 104-112, but especially the last sentence about “halo” effect.
- Acronyms should not be used, such as TFP, line 131.
- What is a “constant comparative method qualitative research approach,” line 195. Each such construct needs to be better defined and described.

--> We corrected according to these helpful hints.

- It should be indicated, if correct, that “psychodynamic” and “psychoanalytic” are being used interchangeably.

--> This also has been made clearer, also adding some additional references.

- The paragraph starting on Line 197 does not describe who made those decisions.

--> We added this.

The tables are not clear where each factor starts and ends.

--> The table is clearer now. Many thanks!

Reviewer 2 Report

Very important contribution to the literature - I was honored to review your work. Well investigated. Well written.

Recommend review for grammar and punctuation (some minor grammar errors, e.g., comma use, spelling mistakes).

Recommend inclusion of more specific information regarding findings on construct validity.

Any future plans to conduct a CFA? To edit the times that didn't "hang well" with the others?

Author Response

Open Review

English language and style

( ) Extensive editing of English language and style required 
( ) Moderate English changes required 
(x) English language and style are fine/minor spell check required 
( ) I don't feel qualified to judge about the English language and style 

Yes

Can be improved

Must be improved

Not applicable

Does the introduction provide sufficient background and include all relevant references?

(x)

( )

( )

( )

Is the research design appropriate?

(x)

( )

( )

( )

Are the methods adequately described?

(x)

( )

( )

( )

Are the results clearly presented?

(x)

( )

( )

( )

Are the conclusions supported by the results?

(x)

( )

( )

( )

Comments and Suggestions for Authors

Very important contribution to the literature - I was honored to review your work. Well investigated. Well written.

Recommend review for grammar and punctuation (some minor grammar errors, e.g., comma use, spelling mistakes).

We corrected for spelling and stile.

Recommend inclusion of more specific information regarding findings on construct validity.

Thank you for the hint, we currently plan to conduct calculations on construct validity within a larger sample. The limitation has been added referring to the small sample size.

Any future plans to conduct a CFA? To edit the times that didn't "hang well" with the others?

We added our current activity concerning the CFA, which was planned, currently carried out and discussed already.
